



# Satellite-based remote sensing data set of global surface water storage change from 1992 to 2018

Riccardo Tortini[1], Nina Noujdina[1], Samantha Yeo[1], Martina Ricko[2], Charon M. Birkett[3], Ankush Khandelwal[4], Vipin Kumar[4], Miriam E. Marlier[5], Dennis P. Lettenmaier[1]

[1]Department of Geography, University of California - Los Angeles, Los Angeles, CA, USA
[2]KBRwyle Inc., Greenbelt, MD, USA
[3]NASA Goddard Space Flight Center, Greenbelt, MD, USA
[4]Department of Computer Science and Engineering, University of Minnesota, Minneapolis, MN, USA
[6]Institute of the Environment and Sustainability, University of California - Los Angeles, Los Angeles, CA, USA

*Correspondence to*: Riccardo Tortini (rtortini@ucla.edu)

**Abstract.** The recent availability of freely and openly available satellite remote sensing products has enabled the implementation of global surface water monitoring to a level not previously possible. Here we present a global set of satellite-derived time series of surface water storage variations for lakes and reservoirs for a period that covers the satellite altimetry era. Our goal is to promote the use of satellite-

derived products for the study of large inland water bodies, and to set the stage for the expected availability of products from the Surface Water and Ocean Topography (SWOT) mission, which will vastly expand the spatial coverage of such products, expected from 2021 on. Our general strategy is to estimate global surface water storage changes ($\Delta V$) in large lakes and reservoirs using a combination of paired water surface elevation (WSE) and water surface area (WSA) extent products. Specifically, we

use data produced by multiple satellite altimetry missions (TOPEX-Poseidon, Jason-1, Jason-2, Jason-3, and ENVISAT) from 1992 on, with surface extent estimated from Terra/Aqua Moderate Resolution Imaging Spectroradiometer (MODIS) from 2000 on. We leverage from relationships between elevation and surface area (i.e., hypsometry) to produce estimates of $\Delta V$ even during periods when either of the variables was not available. This approach is successful provided that there are strong relationships

between the two variables during an overlapping period. Our target is to produce time series of $\Delta V$ as well as WSE and WSA for a set of 347 lakes and reservoirs globally for the 1992-2018 period. The data sets presented are publicly available and distributed via NASA's Jet Propulsion Laboratory's Physical Oceanography Distributed Active Archive Center (PO DAAC; https://podaac.jpl.nasa.gov/). Specifically, the WSE data set is available at https://doi.org/10.5067/UCLRS-GREV2 (Birkett et al.,

2019), the WSA data set is available at https://doi.org/10.5067/UCLRS-AREV2 (Khandelwal and Kumar, 2019), and the $\Delta V$ data set is available at https://doi.org/10.5067/UCLRS-STOV2 (Tortini et al., 2019). The records we describe represent the most complete global surface water time series available from the launch of TOPEX-Poseidon in 1992 (beginning of the satellite altimetry era) to near-present. The production of long-term, consistent, and calibrated records of surface water cycle variables such as

the data set presented here is of fundamental importance to baseline future SWOT products.



## 1 Introduction

Information about surface water dynamics is required to support monitoring and reporting programs
associated with water management as well as scientific objectives such as understanding the space-time
variability of water stored at or near the land surface (Lettenmaier and Famiglietti, 2006). However,
surface water storage data are scarce and often inaccessible in many regions of the world due to
geographic remoteness and/or closed data policies, in addition to the costs associated with maintaining
extensive water monitoring programs. This is especially the case in areas with sparse populations and in
the developing world, limiting our ability to understand the surface water balance at the global scale,
and therefore its effect on water management planning, global weather forecasting, ecosystem
sustainability, and earth system modeling in general (Gao, 2015). The synoptic nature of satellite-based
remote sensing platforms make them ideally suited to quantitatively capture and portray conditions over
large areas at a given point in time, and to characterize how these conditions change through time over
long periods (Lettenmaier et al., 2015; Crétaux et al., 2016). With the recent availability of free and
open access satellite remote sensing products, users now have access to high-quality, analysis-ready
imagery at spatial resolutions that are informative at the relevant scales of variation of WSE and WSA,
and ultimately storage, at least for relatively large inland water bodies. As a result, in recent years the
hydrology community has been active in developing approaches to enable the implementation of global
surface water monitoring strategies (McCabe et al., 2017). Global water dynamics studies that
previously would have only been approachable with relatively low spatial resolution data sets or
gravimetric remote sensing such as GRACE (e.g., Humphrey et al., 2016) are now implemented using
high resolution imagery such as Landsat. For example, the European Commission Joint Research
Center's Global Surface Water Explorer quantifies changes in global surface water at 30 m resolution
for a 32-year period (Pekel et al., 2016). In addition, despite being primarily designed to measure water
levels over the open ocean, current generation satellite altimetry missions have demonstrated their
suitability for hydrological studies for large inland water bodies, both for specific targets such as Lake
Chad (Coe and Birkett, 2005), the Aral Sea (Aladin et al., 2005; Singh et al., 2012), and at the regional
scale, for example the African Great Rift Valley Lakes (Birkett et al., 1999), and the Tibetan Plateau
(Lee et al., 2011; Kleinherenbrink et al., 2015; Cai et al., 2016). Extensive efforts have been made to
measure surface height for large lakes and reservoirs globally; examples include the French Space
Agency - Laboratoire d'Etudes en Géophysique et Océanographie Spatiales hydroweb database
(LEGOS; Crétaux et al., 2011), the Database for Hydrological Time Series of Inland Waters (DAHITI;
Schwatke et al., 2015), and the U.S. Department of Agriculture (USDA) Global Reservoir And Lake
Monitoring (G-REALM) data sets. However, surface water storage estimation at the global scale
remains challenging and still largely unexplored (Gao et al., 2012; Gao, 2015). NASA's upcoming
Surface Water and Ocean Topography (SWOT) mission (scheduled launch 2021) will fill a major void
in the global observational capabilities of the hydrology community. SWOT is expected to produce
accurate WSE and WSA estimates on average every 10.5 days (depending on specific location) with the
ability to estimate surface water storage variations for lakes and reservoirs as small as about 1 km$^2$ with
a height accuracy of around 10 cm (Biancamaria et al., 2010). However, until SWOT data are available,
the development of satellite-based long-term hydrologic records for the study of variability and changes
in the terrestrial water cycle will demand accurate data homogenization and harmonization from



existing sensors, with transparent and reproducible methods playing a pivotal role to obtain consistent and defensible results (McCabe et al., 2017). Moreover, given that the current generation of altimeters are nadir-pointing, i.e., provide information along tracks rather than swaths (typically with track separation order of 100 km or so), long-term records can be obtained exclusively by merging data sets

from a constellation of sensors with a range of (often overlapping) data records. For example, Crétaux et al. (2016) estimated that the constellation of Jason-2, Jason-3, France-India SARAL/AltiKa (Verron et al., 2015), and European Space Agency's Sentinel-3A/3B tandem (Donlon et al., 2012) has the potential to capture water surface elevation (WSE) for nearly the entirety of 3,720 global lakes with areas larger than 50 km$^2$ and 71% of the 14,411 lakes larger than 10 km$^2$, for a total of approximately

40% of the global water storage of lakes on Earth. However, this merging of records from heterogeneous satellite sources has practical drawbacks such as discontinuities in the derived water storage estimates, and the harmonization of these sources is fundamental to achieving more effective data assimilation for use in, for example, hydrological models, with the direct consequence of triggering a better understanding of any underlying physical process (McCabe et al., 2017). Here we summarize

results of the integration of long-term satellite remote sensing data collected by optical and microwave sensors to produce global surface water storage records for large lakes and reservoirs, beginning with the launch of TOPEX/Poseidon (T/P) in 1992. We use data produced by multiple satellite altimetry missions, including but not limited to T/P, Jason-1, Jason-2, and Jason-3, with surface extent estimated from MODIS from 2000 on. We leverage from the relationship between WSE and WSA (i.e.,

hypsometry) to produce estimates of storage changes ($\Delta V$) even during periods when either of the variables are not available, as long as there are strong relationships between the two during an overlapping period. If the correlation coefficient between the two variables was smaller than 0.85 and the variance of either variable was smaller than 2%, we simplified the model into a single variable (i.e., noninvariant) function. Our intent is to produce the most complete possible satellite-derived records of

water $\Delta V$ over the period from the T/P launch up to the launch of the SWOT mission, with the goal of providing long-term, consistent, and calibrated records of baseline surface water cycle variables up to the time of SWOT launch and beyond.

## 2 Data and methods

In this section, we describe the remote sensing data sources and the methods we used to estimate WSE,

WSA, and $\Delta V$. Given the technological limitations of the currently operational satellite platforms we used, we targeted water bodies globally with (i) WSE time series overlapping with WSA time series so that a hypsometric curve could be established for the 2000-2016 period; (ii) reference WSAs larger than 30 km$^2$ (approximately 120 MODIS pixels with 500 m resolution); and (iii) lakes or reservoirs that were clearly distinguishable from other nearby water bodies (improved accuracy of both WSE and WSA

estimates). As an example of the records we analyzed and their capabilities, we perform a detailed analysis of Lake Sakakawea (47.50°N; 101.41°W), a large reservoir located in the Missouri River Basin in the Fort Berthold Indian Reservation in central North Dakota (USA) and impounded by the Garrison Dam. Figure 1 shows the location of the lakes and reservoirs selected for this work, with a close up of Lake Sakakawea.

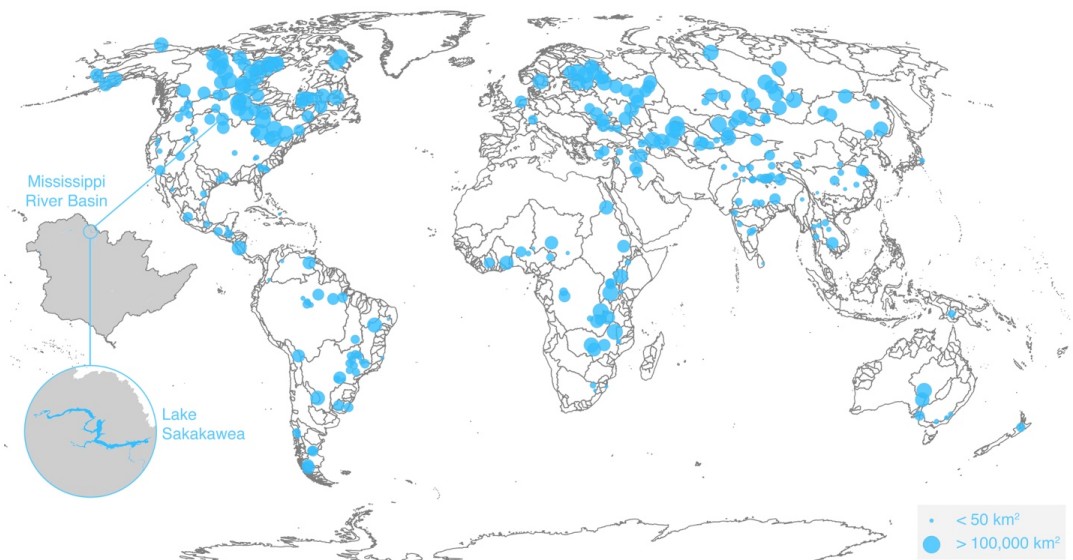

**Figure 1: Location of the global targets (blue bubbles, by average lake size) and Lake Sakakawea (approximate coordinates: 47.50°N; 101.41°W) within the Mississippi River Basin (shaded).**

### 2.1 Water surface elevation

G-REALM10 merges T/P, Jason-1, Jason-2, and Jason-3 time series of relative WSE variations with respect to a given Jason-2 reference cycle at 10-day intervals (Birkett, 1995; Birkett and Beckley, 2010; Birkett et al., 2011), whereas, whenever 10-day measurements are not available, G-REALM35 is

10 created using the ENVISAT time series of relative water level variations, for which the mean level of ENVISAT retrievals at 35-day intervals is the reference. $\Delta V$ monitoring of inland water bodies at the global scale has proved a challenging task (Gao et al., 2015; Crétaux et al., 2016), and the use of a single WSE data source significantly limits the creation of global $\Delta V$ data set. For these reasons, we used G-REALM10 as our primary elevation source for the creation of our global $\Delta V$ data set, and,

whenever G-REALM10 was not available for a specific target, supplemented it with LEGOS, DAHITI, and G-REALM35 (in this order) based on factors such as density and trend of the available measurements.

Figure 2 shows the radar altimeter ground tracks over Lake Sakakawea, where we merged multiple data sources to create the G-REALM10 and G-REALM35 records. We extracted WSE data for the portions

of the ground tracks over the water body and used them to construct a time series of WSE variations. We used 10-day records from the TOPEX/Poseidon and Jason instrument series (1992-2002, and 2008-2017) with 35-day ENVISAT mission data used during the 2002-2008 period. A more detailed description of the methods we used can be found in Birkett (1995), Birkett and Beckley (2010), and



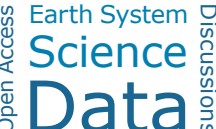

Birkett et al. (2011). Ricko et al. (2012) performed both absolute and relative validations between the various G-REALM, DAHITI and LEGOS available product types and for the majority found an acceptable level of accuracy between them.

WSE accuracy is highly affected by the presence of ice, and for practical purposes, reliable $\Delta V$
estimates can only be produced during ice free conditions. We assessed ice-on conditions (i.e., presence of snow-covered ice on the surface of a water body) using the MODIS/Terra Snow Cover Daily Global product (Collection 5 MOD10A1). For each elevation record, we estimated lake ice phenology (i.e., ice-on and ice-off dates, defined as the beginning and end of the freezing period) as the proportion of frozen pixels identified in the NDSI-based 500 m spatial resolution "Snow_Cover_Daily_Tile" band (Hall et
al., 2007), and we determined a threshold for each water body as half of the maximum observed WSA. This algorithm uses the basic assumption that a water body, when deep and clear, absorbs the solar radiation incident upon it in almost its entirety. Whenever ice was identified, we created a flag that is provided as part of the $\Delta V$ records. Water bodies with high turbidity, algal blooms, or other conditions of relatively high reflectance from the water (e.g., salt crust) may be erroneously detected as snow
and/or ice covered; in these cases we manually removed the ice flag. We classified data gaps within the freezing period as ice-on for continuity purposes. Additionally, we excluded observations during polar darkness for lack of complete data and likely ice-on.

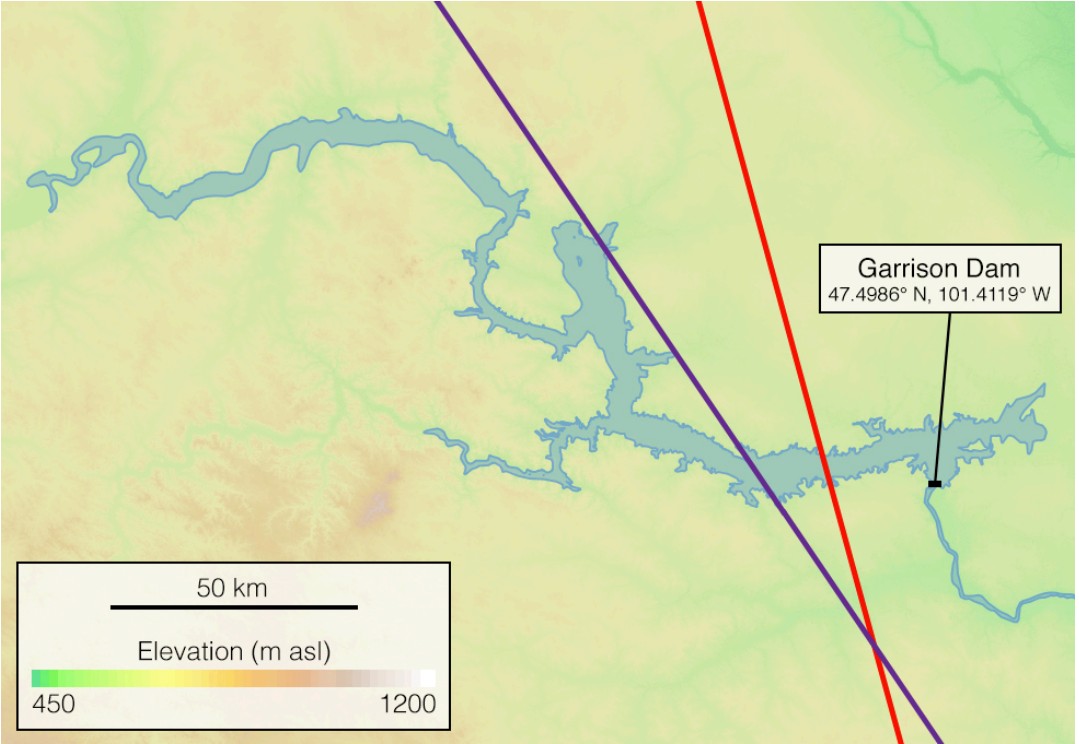





**Figure 2: Radar altimeter ground tracks over Lake Sakakawea (blue) overlaid to the SRTM 1-arc digital terrain model. Purple: 10-day resolution instrument series and satellite pass 204; red: 35-day resolution series and satellite pass 323.**

**2.2 Surface water area**

The Global Optical Lake Area (GOLA) determination process estimates WSA of lakes and reservoirs from Terra/Aqua MODIS satellite optical imagery with a 500 m spatial resolution and an 8-day temporal resolution for the 2000-2016 period. In order to estimate the WSA of the target, a static spatial extent is required as one of the inputs (Khandelwal et al., 2017). We defined the initial spatial extents of water bodies using the vector polygons available as part of the Global Reservoir and Dam Database

(GRanD; Lehner et al., 2011) and Global Lakes and Wetlands Database (GLWD; Lehner and Döll, 2004), with quality checks ensured by visual comparison with high resolution satellite imagery (i.e., Google Earth, ESRI World Map). Whenever we identified a mismatch (i.e., polygon spatial extent not overlapping properly with the satellite imagery due to inaccurate georeferencing), the polygon was edited to match the expected location. In case a water body was not available as part of either database,

a polygon was drawn by hand using high resolution imagery from various sources (e.g., Global Surface Water Explorer, Google Earth, ESRI World Map). Once correctly identified, these locations were used to construct a mask for MODIS data extraction. We then used the mask to extract all of the data from three MODIS products whose nominal footprint overlapped the polygon of the corresponding lake. Specifically, we used: (i) two multispectral reflectance data products from the MODIS instruments

onboard NASA's Terra and Aqua satellites as an input to the water/land classification algorithm (Collection 5 MCD43A4 and MOD0911), and (ii) static water and land classification labels to train the classification model (MODIS MOD44W). The primary reflectance product was the bidirectional reflectance distribution function (BRDF) adjusted MCD43A4 16-day composite product. The MCD43A4 product is generated by the U.S. Geological Survey (USGS) using data from both the Terra

and Aqua satellites to assure that the combined data product is of the highest possible quality. However, by ignoring poor data quality pixels, the MCD43A4 product suffers from a high degree of missing values, especially before Aqua data became available in 2002. This can introduce a high degree of incompleteness in classification maps. To alleviate this issue, we also used the MOD09A1 8-day composite product collected solely from the Terra satellite. Since the MOD09A1 product is generally

less reliable than MCD43A4 as it is not BRDF-adjusted, we combined these two products to compensate for the primary limitations of each, in addition to noise and missing values following methods outlined by Khandelwal et al. (2017). We also used quality flags to filter out pixels with snow, ice, or clouds. For the MOD10A1 product, information about the data quality is available along with the multispectral values in the 16-bit quality assessment state flags, whereas the quality flags for the

MCD43A4 product are available as a separate product (MCD43A2 BRDF/Albedo Quality Product). In order to distinguish between land and water bodies, we used static water extent masks derived from the MODIS MOD44W product (Carroll et al., 2009) to train the supervised classification models. This product, distributed publicly by the USGS, combines MODIS 250 m reflectance data with the SRTM Water Body Dataset from 60°N to 60°S, with reflectance data used solely poleward of 60°N. We

aggregated the MOD44W product from 250 m to 500 m to match the resolution of the other MODIS

products. In particular, if the 500 m pixel had all of its four pixels at 250 m labeled as water or land in the MOD44W product, then we considered the pixel as a water or land pixel. We excluded partial pixels from the training set pool. Figure 3 shows an example of the classification results for Lake Sakakawea under a dry and a wet scenario. A more detailed description of the classification algorithm and its
validation can be found in Khandelwal et al. (2017). All MODIS data used to create the GOLA records are publicly available via the USGS Land Processes Distributed Active Archive Center (LP DAAC; http://lpdaac.usgs.gov).

### Lake Sakakawea - GOLA water surface area classification



**Figure 3: Examples of the GOLA WSA classification results for Lake Sakakawea: (a) dry scenario (November 1st, 2008); (b) wet scenario (April 25th, 2011). Differences in WSA estimates are noticeable in the northwestern and southwestern branches of the reservoir, the farthermost from the Garrison Dam.**

### 2.3 Global storage change

During time periods when both WSEs from G-REALM (supplemented with DAHITI and LEGOS) and WSAs from GOLA were available, we derived the elevation-surface area relationships (i.e., hypsometry) for each target. We then used these relationships to estimate reservoir $\Delta V$ using an approach similar to Gao et al. (2012). Specifically, for overlapping G-REALM and GOLA periods, we calculated increments of volume for the corresponding changes in WSE and WSA as:


$$\Delta V = (WSA_{t+1} + WSA_t)(WSE_{t+1} + WSE_t)/2,$$
$$(1)$$

where $WSA_t$ and $WSE_t$ are surface area and elevation at the smallest step $t$, and $A_{t+1}$ and $h_{t+1}$ are surface
area and elevation at the next incremental step $t+1$.
    We used linear regression to approximate the relationship between elevation ($WSE$) and surface area ($WSA$), $WSA = f(WSE)$. We then applied this relationship to estimate WSA from WSE for periods when WSA is unavailable (1992-1999), and the inverse function $WSE = f^{-1}(WSA)$ to estimate WSE from





WSA for periods when WSE is unavailable during the MODIS era (2017-2018). Finally, the $\Delta V$ equation can be simplified into a single variable function, either as a function of WSE or GOLA WSA, by substituting $WSA = f(WSE)$ or $WSE = f^{-1}(WSA)$ into it. If the correlation coefficient between the two variables was smaller than 0.85 (i.e., weak to moderate correlation between WSE and WSA) and the 5 variance of either variable was smaller than 2% (i.e., near-invariant variable), then we parameterized the invariant variable using its mean value.

### 3 Results

We created water storage records for 347 global lakes and reservoirs, distributed via NASA's Jet Propulsion Laboratory's Physical Oceanography Distributed Active Archive Center (PO DAAC; 10 https://podaac.jpl.nasa.gov/). Table 1 summarizes WSE, WSA, and $\Delta V$ per continent of the water bodies with records in the period of this work (i.e., 1992-2018). The majority of the water bodies are located in Asia and North America (223, 64.26%), with Australasia represented by just eight targets. Globally, approximately 22% of the WSE measurements overlap with WSA records enabling hypsometric curves to be constructed, with no significant regional exception. Africa and North America lead in terms of 15 average WSA, with an average of ~4864 km² (39 water bodies) and ~4100 km² (113 water bodies), respectively. In fact, the dynamics of the water bodies in Africa are dominated by the Great Rift Valley Lakes, whereas the size range of the water bodies in North America is more varied. South American water bodies instead show the highest variability (i.e., standard deviation) per average area (118.47 km² and 1072.33 km², respectively), compatible with the generally modest topographic relief and frequent 20 flooding of the major rivers and reservoirs. However, Africa also has the largest observed mean decrease in both $\Delta V$ (-377.74 km³) and standard deviation (3.77 km³), suggesting shallow topography and highly dynamic variations.

**Table 1: Summary by continent of the observed characteristics of the 347 water bodies.**

25

| Continent | Water bodies | Average per target | | WSE [m] | | WSA [km²] | | $\Delta V$ [km³] | |
| | | Water level records | Hypsometric records | Mean | Standard deviation | Mean | Standard deviation | Total | Standard deviation |
|---|---|---|---|---|---|---|---|---|---|
| Africa | 39 | 378.87 | 237.61 | -0.62 | 1.87 | 4864.36 | 100.12 | -377.74 | 3.77 |
| Asia | 110 | 361.84 | 187.63 | -1.22 | 3.61 | 1736.74 | 114.45 | -171.86 | 2.40 |
| Australasia | 8 | 231.00 | 179.62 | -0.98 | 3.97 | 385.85 | 43.34 | -159.76 | 0.60 |
| Europe | 28 | 554.11 | 236.07 | +0.06 | 0.59 | 2665.49 | 98.91 | -116.67 | 1.35 |
| North Am. | 113 | 458.44 | 169.85 | -0.34 | 1.67 | 4099.97 | 65.34 | -115.01 | 1.92 |
| South Am. | 49 | 291.84 | 178.08 | -0.43 | 2.60 | 1072.33 | 118.47 | -120.73 | 1.91 |
| Global | 347 | 379.35 | 198.14 | -0.59 | 2.38 | 2470.79 | 90.10 | -176.96 | 1.99 |

Figure 4 shows the monthly frequency of the observations used to create the hypsometric curve for the 347 targets we analyzed. The total number of hypsometric observations was 65,872 (average 30 observations per target: 189.83, or ~11 per overlapping year). With the majority of the targets located in the Northern Hemisphere (272 targets, 78.4% of the total), 55.86% of the total hypsometric records are observed in the Boreal late spring and summer months (May-September) and only 26.76% in the Boreal

late fall and winter (November-March), due to a combination of factors such as fewer optical images with cloud cover, absence of ice cover, and in general more accurate WSE estimates.

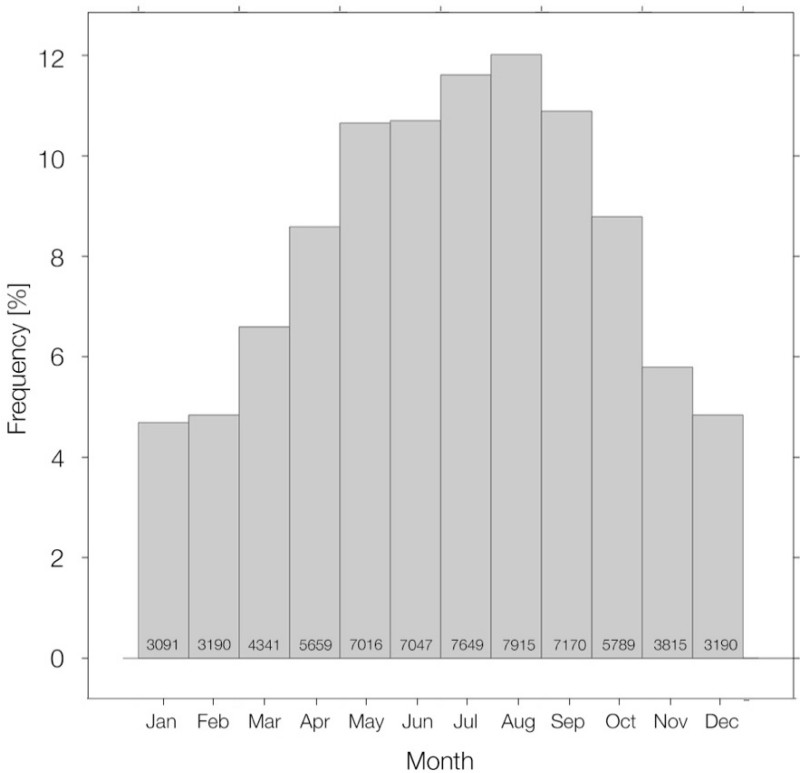

**Figure 4: Monthly frequency of the observations used to create the hypsometric curve for the 347 targets analyzed in this study, with total number of observations for each month.**

Figure 5 shows the temporal trends of the observed G-REALM elevation and GOLA surface area records for Lake Sakakawea. Both data sets show consistent trends and seasonal variations for the
overlapping period (2000-2016). The smoother seasonality associated with the GOLA records may be a direct consequence of the spectral heterogeneity associated with the low spatial resolution (i.e., 500 m) of the pixels along the target boundary. In addition, the sparser G-REALM35 records only partially compensate for the unavailability of G-REALM10 records from 2003 to 2008 (Figure 5a). However, the denser GOLA time series in the same period (Figure 5b) offers the potential to supplement further $\Delta V$
records based on the observed relationship with elevation records. This is especially relevant because the drainage area to Lake Sakakawea suffered a significant drought in the early 2000s. In fact, by May 2005 Lake Sakakawea had fallen to a documented all-time low of 1,805.8 ft msl (~550.4 m; US Army Corps of Engineers, 2007). However, thanks to a wet early summer in 2008 and the spring runoff of



2009, by 2010 Lake Sakakawea was nearly at full capacity. These dynamics are reflected in both the G-REALM and GOLA records (Figure 5) and are consistent with the results obtained by Gao et al. (2012).

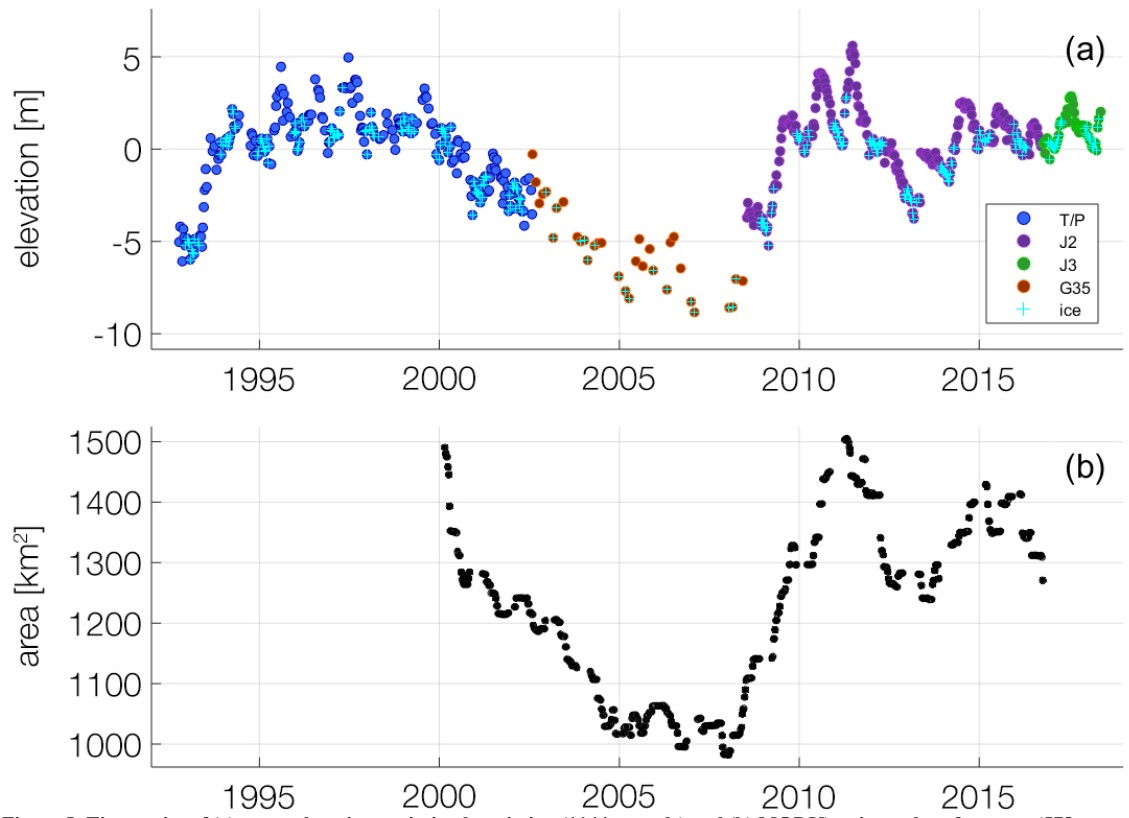

**Figure 5: Time series of (a) water elevation variation by mission (1144 records) and (b) MODIS-estimated surface area (578 records) for Lake Sakakawea. Presence of surface ice is indicated by a light blue cross.**

Figure 6 shows the hypsometric curve for Lake Sakakawea (R = 0.908). Such a high correlation usually indicates good quality for both data sets; conversely, low correlations can result from many conditions.
These include systematic errors in either water elevation or surface area records (or both), and/or geomorphic properties of the target, with the possibility that, within the range of variation of either variable, the hypsometry is more or less independent of surface area (i.e., in the extreme vertical walls) or elevation (i.e., shallow lakes). Whenever direct observations of WSE were unavailable, we used the hypsometric curve to derive two associated products: inferred water elevation records and inferred
surface area records.

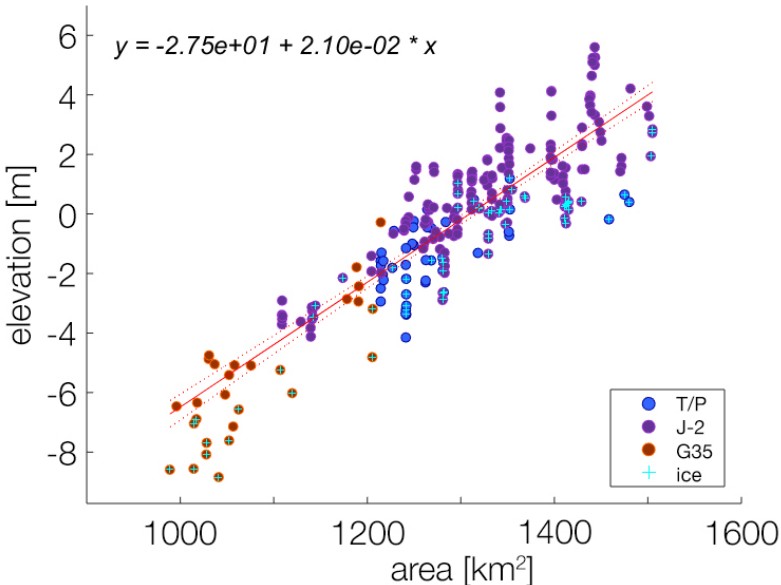

$y = -2.75e+01 + 2.10e-02 * x$

**Figure 6: Water elevation and surface area relationship for Lake Sakakawea (277 records).**

For the overlapping period (2000-2016) when both WSE and WSA were available, G-REALM was
5   used in the final product to compute the relative storage because of its more relevant role played in
modelling of $\Delta V$ (cfr. Eq. (1)). Figure 7 shows the estimated relative storage time series for Lake
Sakakawea.

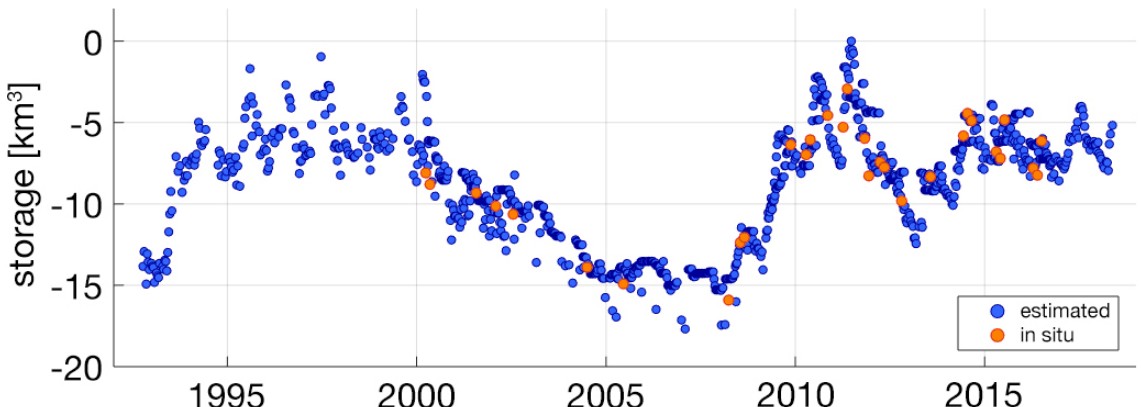

10   **Figure 7: Time series of relative storage for Lake Sakakawea. Observed records are in black, modelled records are in blue.**

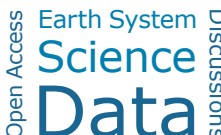

**4 Validation**

We evaluated the statistical accuracy of WSE and storage estimates at Lake Sakakawea based on monthly in situ water measurements made by the U.S. Army Corps of Engineers at Garrison Dam (http://www.nwd-mr.usace.army.mil/rcc/projdata/garr.pdf) and available until from June 1967 to

5   December 2018 (Fig. 8a-b). Specifically, we utilized the "Average Daily Midnight Elevation (ft msl)" and "End-of-Month Storage (1,000 AF)" products. After averaging the WSE records to the monthly scale, 233 and 270 coincident observations were available for WSE and storage change, respectively. The RMSE of the WSE was ~0.68 m. The linear fit had an $R^2 = 0.95$ ($p < 0.001$), suggesting very good consistency of the in situ water level measurements and the derived optical water levels (Figure 8c). The

10   RMSE of the storage change was 0.87 km$^3$. The linear fit had an $R^2$ of 0.94 ($p < 0.001$), indicating very good consistency with the in situ storage estimates (Figure 8d).

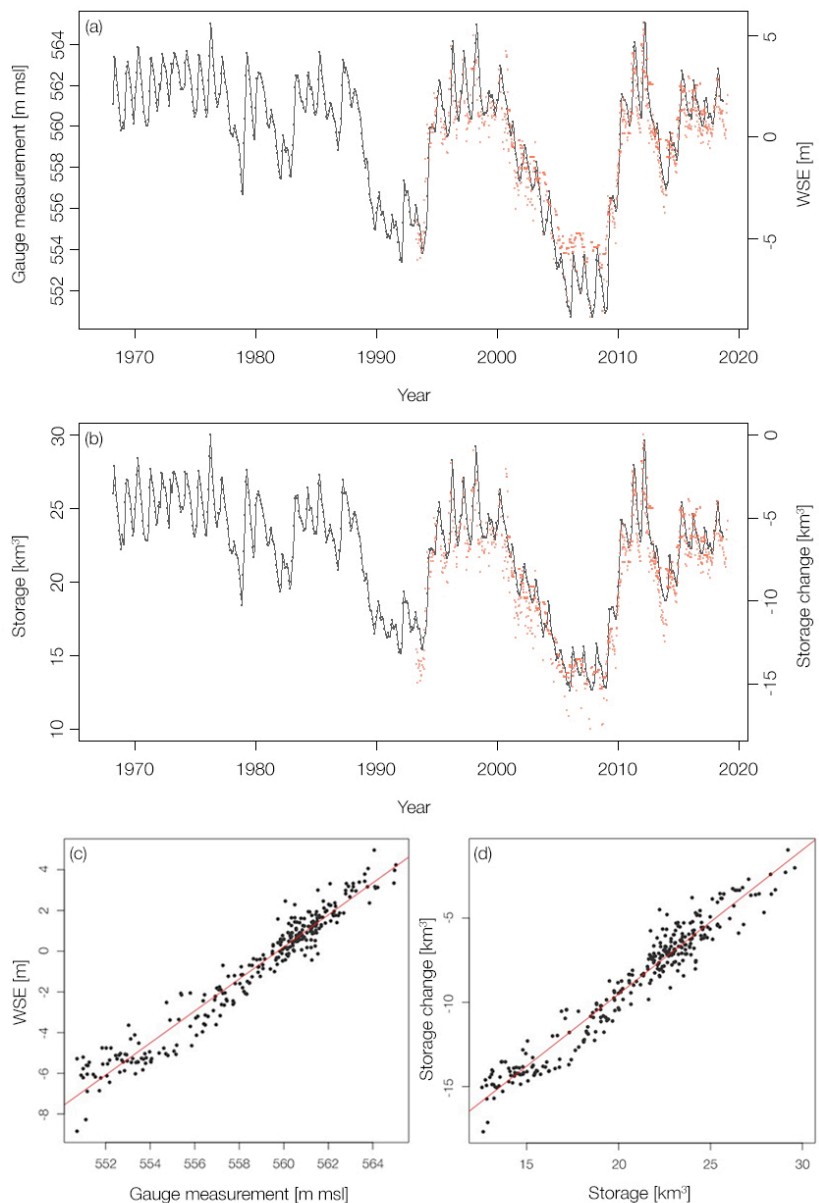

**Figure 8: Water levels and storage at Lake Sakakawea. (a) In situ monthly water levels (black) versus WSE records (red); (b) in situ monthly water storage (black) versus *ΔV* records (red); (c) linear regression of monthly average WSE records and concurrent in situ monthly water levels, with linear regression in red; (d) linear regression of monthly average *ΔV* records and concurrent in situ monthly water storage, with linear regression in red.**

**5 Discussion**

In the Lake Sakakawea example, both the G-REALM and GOLA records show consistent trends and seasonal variations for the overlapping period (2000-2016). Inaccuracy in the estimated relative storage
can be attributed mainly to (i) WSE errors, (ii) WSA errors, and (iii) WSE-WSA relationship errors. The accuracy of the elevation records can be attributed to a number of factors, including satellite orbit, distance between antenna and target (i.e., altimetric range), geophysical range corrections, target size, and track location relative to the target boundary. Furthermore, each WSE record is calculated as the average value along the satellite ground track, with a large standard error implying higher uncertainty
potentially from both measurement errors and/or natural variations (e.g., surface roughness). For example, satellite tracks over narrow water bodies in complicated terrain will result in larger errors. Finally, major wind and precipitation events, as well as tidal effects and the presence of ice also affect the quality of the records. The spectral heterogeneity associated with pixels along the target boundary plays a key role in the accuracy of the surface area classification. For example, Lake Sakakawea is a
sinuous water body of 286 km length at capacity and average width of 3-5 km. As a result, a significant number of the MODIS 500 m pixels used to analyze the target are spectrally heterogeneous (i.e., partially covered by water and land) and therefore more prone to misclassification. This is especially true for droughts and/or periods of low water levels, as sinuous water bodies become even narrower due to drying. In addition, targets with limited or near-static water dynamics (defined as "dynamic region
width" by Khandelwal et al., 2017) present land cover changes in the GOLA product primarily near the boundary of the static region used in the classification. Due to the moderate spatial resolution of the GOLA records, the effect of mixed pixels is even more prominent in water bodies with low dynamic region width, which can lead to low correlation values between elevation and surface area. Conversely, the classification of targets with high dynamic region width consistently performs better in the GOLA
records. The quality of both elevation and surface area contribute to the accuracy of their relationship. High correlations between elevation and area generally indicate reliable $\Delta V$ estimation. However, if either variable is systematically biased, the error associated with the relationship is carried to the estimated $\Delta V$. For example, low correlation may arise when the target shows nearly constant WSA (vertical walls, in which case a variation in elevation reflects in a negligible change in WSA) or nearly
constant elevation (i.e., shallow lakes, in which case a variation in surface area reflects in a negligible change in elevation). In these cases we proceeded in the modelling of $\Delta V$ with the parameterization of the invariant variable with its mean value. All the factors listed above introduce some degree of error in the WSE-WSA relationship; however, in most cases a linear approximation does not appear to be a major contributor (cfr. Gao et al., 2012). At the global scale, the limited number of altimeter-based
WSE products is a key constraint for satellite remote sensing observations. In fact, due to the technical limitations listed above, current generation spaceborne microwave altimeters can only monitor WSEs for a relatively small number of large reservoirs when used individually. In order to maximize the length and density of global $\Delta V$ records, in addition to integrating measurements from multiple altimeters, multiple MODIS daily overpasses played a crucial role in creating consistent 8-day GOLA
and consequently $\Delta V$ records.





Despite GOLA's moderate spatial resolution it can potentially affect the accuracy of $\Delta V$ estimates, higher resolution satellite missions have longer satellite revisit time (e.g., 16 days for Landsat). Because we leveraged the relationship between WSE and WSA to estimate $\Delta V$, such satellite revisit times would produce sparser records, especially for water bodies located at high latitudes and/or altitudes as they are more affected by cloud cover. In fact, despite being highly desirable for monitoring of surface water dynamics, imagery from optical sensors is strongly affected by the presence of cloud cover, which can be extensive in late fall and winter, and in combination with low sun angle experienced at high latitudes may limit its usefulness at the global scale (Duguay et al., 2015). However, the integration of optical imagery (e.g., MODIS, Landsat) and radar altimetry data provides long-term continuity in the production of consistent and calibrated records.

## 6 Data availability

The data sets presented are publicly available and distributed via NASA's Jet Propulsion Laboratory's Physical Oceanography Distributed Active Archive Center (PO DAAC; https://podaac.jpl.nasa.gov/). Specifically, the WSE data set is available at https://doi.org/10.5067/UCLRS-GREV2 (Birkett et al., 2019), the WSA data set is available at https://doi.org/10.5067/UCLRS-AREV2 (Khandelwal and Kumar, 2019), and the $\Delta V$ data set is available at https://doi.org/10.5067/UCLRS-STOV2 (Tortini et al., 2019).

## 7 Summary

We generated global water storage change ($\Delta V$) estimates based exclusively on satellite remote sensing observations through the creation of elevation (i.e., G-REALM) and surface area (i.e., GOLA) associated products for 347 selected large water bodies, primarily based on the availability of water elevation products. G-REALM10 was derived from a constellation of satellite altimeters (i.e., TOPEX/Poseidon, Jason-1, Jason-2, Jason-3), whereas G-REALM35 was created using measurements from ENVISAT. We supplemented the G-REALM elevation records with DAHITI and LEGOS products. We utilized the algorithm developed by Khandelwal et al. (2017) to create 8-day 500 m surface area estimates from MODIS images. WSE and WSA were used to derive the hypsometric relationship for each reservoir, with either variable inferable from its counterpart when direct observations were unavailable. We computed $\Delta V$ using an adaptation of the method of Gao et al. (2012). As an example, we demonstrate application of the data set to Lake Sakakawea (North Dakota, USA), the second largest reservoir in the USA by area, and representative of the challenges encountered in the creation of global $\Delta V$ records. The records presented in this paper represent the most complete satellite-derived global surface water storage time series to date, spanning from 1992 (TOPEX-Poseidon launch) to present, with the potential to be extended up to the launch of the SWOT mission planned for 2021. The data set presented is dynamic and will continue to be extended both in terms of the number of water bodies (with ultimate potential total around 400), and historical time period. Despite the coarser spatial resolution of the pre-SWOT records presented, the production of long-term, consistent, and





calibrated records of surface water cycle variables is of fundamental importance to establishing a
baseline of what is known globally about surface water $\Delta V$ up to the time of SWOT launch.

**Acknowledgements**

The data set presented is available as additional material to this paper and distributed via NASA's PO
DAAC (https://podaac-tools.jpl.nasa.gov/drive/files/allData/preswot_hydrology/) as L2 (level
variation), L3 (surface area), and L4 (storage change) products. This work was funded by NASA
Making Earth System Data Records for Use in Research Environments (MEaSUREs), Grant No
NNX13AK45A to UCLA. Kumar and Khandelwal were supported by NSF Grants #1029711 and
#1838159. We would like to thank Jessica Hausman (NASA JPL) for comments on the content and
format of the records produced in this work, and Jongyoun Kim for her work on earlier versions of the
data set. We acknowledge that Lake Sakakawea lies on the traditional territory of the Mandan, Hidatsa,
and Arikara, who still walk the land today.

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
