# Peer review of "Satellite-based remote sensing data set of global surface water storage change from 1992 to 2018"

_Earth System Science Data, 2019_

## Referee Comment (RC1) · Anonymous Referee #1 · 28 Jan 2020

In this study, the authors produced a global lake/reservoir volume change dataset. The water levels are derived from altimetry data between 1992 and 2018. The water areas are mapped from MODIS data between 2000 and 2018. Finally, the water storage gain or loss for 347 lakes/reservoirs are estimated. This study is suitable to published in ESSD, but some improvements (see comments below) are necessary.

Major comments:

1. The WSA is estimated using 500-m MODIS data. It looks that 120 MODIS pixels are included, but the lake surface area change is used. This is not suitable for most of lakes with small area changes. It could be fine for reservoirs, as the reservoirs have large

inundated area dynamics. Many studies have used lake mapping from 30-m Landsat images, which is better than MODIS data.

2. How many lakes in the Tibetan Plateau are included? The existed studies have reported that about 60 lakes with altimetry data and the corresponding estimates of lake volume variations.

3. The Equation (1) is correct? It is WSEt+1 – WSEt?

4. The linear regression between elevation (WSE) and surface area (WSA) was used. How about polynomial correlation? The authors test them?

5. More validations including lakes in different types and continents can be provided?

6. How water storage change in 1992-2000 without MODIS water mapping was estimated?

Specific comments:

1. "and to characterize how these conditions change through time over long periods (Lettenmaier et al., 2015; Crétaux et al., 2016)" A suggested study here for monitoring lake area, level and volume changes since 1970s: http://dx.doi.org/10.1002/2017GL073773

2. "and the Tibetan Plateau (Lee et al., 2011; Kleinherenbrink et al., 2015; Cai et al., 2016)." It looks some key studies of lake changes in the Tibetan Plateau from altimetry data are missed here. https://doi.org/10.1029/2019GL085032, https://doi.org/10.1016/j.rse.2011.03.005, https://doi.org/10.1007/s10712-016-9362-6

3. "a polygon was drawn by hand using high resolution imagery from various sources (e.g., Global Surface Water Explorer, Google Earth, ESRI World Map)" How to make sure the dates between them are matched.

4. The links below are not accessed. https://doi.org/10.5067/UCLRS-GREV2 https://doi.org/10.5067/UCLRS-AREV2 https://doi.org/10.5067/UCLRS-STOV2

https://podaac.jpl.nasa.gov/

5. The correlation (rˆ2) could be presented in Figures 6, 7, 8.

6. How about the mismatching in Figure 8a-b?

––––––––––––––––––––––––––––––––

---

## Referee Comment (RC2) · Anonymous Referee #2 · 18 Feb 2020

The manuscript entitled "Satellite-based remote sensing data set of global surface water storage change from 1992 to 2018" by Tortini et al. presents estimated global surface water storage changes ( $\Delta V$ ) in large lakes and reservoirs using a combination of paired water surface elevation (WSE) and water surface area (WSA) extent products. In their approach, they used data produced by multiple satellite altimetry missions (TOPEX-Poseidon, Jason-1, Jason-2, Jason-3, and ENVISAT) from 1992 on, with surface extent estimated from Terra/Aqua Moderate Resolution Imaging Spectroradiometer (MODIS) from 2000 on.

They used the relationships between elevation and surface area to produce estimates

of $\Delta V$ even during periods when either of the variables was not available. They produce time series of $\Delta V$ as well as WSE and WSA for a set of 347 lakes and reservoirs globally for the 1992–2018 period.

In general I find the idea of manuscript very interesting and I also see the need for having such data base. Indeed, the production of long-term, consistent, and calibrated records of surface water cycle variables such as the data set presented here is of fundamental importance to baseline future SWOT products.

Major comments:

I believe the paper suffers from missing an important point. The authors calculate first the correlation coefficient between the two variables and use it as one of the decision parameter for taking the mean value instead of data itself. The correlation coefficient between two data set represents the linear dependency between two data sets, while the relationship between water level and surface area represent the bathymetry and the bathymetry of a lake should not follow a linear behaviour. I would strongly suggest to change this in the paper and in case the authors would like to assess the monotonic behaviour between water level and area, then they should use the Spearman rank correlation and not simply the Pearson correlation.

My second major comment goes to the methodology for the area extraction. Figure 6 shows some vertical lines of points, which represent same area for different water levels. This is highly suspicious.

Specific comments:

page 2, line 30, please mention River and Lake https://earth.esa.int/web/guest/-/river-and-lake-products-from-altimetry-4617 and HydroSat http://hydrosat.gis.uni-stuttgart.de

page4, Section 2.1, Did you make sure that all data from different data centers have the same background models for atmospheric refraction? How did you deal with the

intersatellite bias?

page 5 line 3, what is an acceptable accuracy? please quantify!

page 7, equation 1, I did not grasp the equation. shouldn't be WSA t+1 - WSA t?

page 7, line 26, considering linear regression is wrong. See my major comment.

Figure 6, the extracted area is so noisy that similar area are obtained for different height. And in fact, no obvious linear relationship can be recognized.

---

## Author Comment (AC1) · 27 Mar 2020

We are thankful to the anonymous reviewer for the thoughtful comments, which led to significant improvements to our manuscript. All comments were addressed as described below.

Best regards,
Riccardo Tortini

[Figure]

**In this study, the authors produced a global lake/reservoir volume change dataset. The water levels are derived from altimetry data between 1992 and 2018. The water areas are mapped from MODIS data between 2000 and 2018. Finally, the water storage gain or loss for 347 lakes/reservoirs are estimated. This study is suitable to published in ESSD, but some improvements (see comments below) are necessary.**

We are thankful to the anonymous reviewer for recommending the publication of our manuscript in ESSD.

**1. The WSA is estimated using 500-m MODIS data. It looks that 120 MODIS pixels are included, but the lake surface area change is used. This is not suitable for most of lakes with small area changes. It could be fine for reservoirs, as the reservoirs have large inundated area dynamics. Many studies have used lake mapping from 30-m Landsat images, which is better than MODIS data.**

We agree with the reviewer on this point. Landsat's finer resolution (i.e. 30 m) compared to MODIS (i.e. 500 m) would ensure the monitoring of smaller lakes, further expanding our list of 347 lakes/reservoirs. However, compared to MODIS, Landsat's 16-day revisit time would not be suitable for dense time series of observations and therefore to establish a robust relationship between WSE and WSA in order to model $\Delta$V. We now emphasize this point in page 16, line 1-10 as reported below.

"Despite GOLA's moderate spatial resolution it can potentially affect the accuracy of $\Delta$V estimates, higher resolution satellite missions have longer satellite revisit time (e.g., 16 days for Landsat, 10 days for Sentinel-2A starting in 2015 and 5 days for Sentinel-2A and -2B in tandem formation starting in 2017). Because we leveraged the relationship between WSE and WSA to estimate $\Delta$V, such satellite revisit times would produce sparser records, especially for water bodies located at high latitudes and/or altitudes as they are more affected by cloud cover. In fact, despite being highly desirable for monitoring of surface water dynamics, imagery from optical sensors is strongly affected by the presence of cloud cover, which can be extensive in late fall and

[Figure]

winter, and in combination with low sun angle experienced at high latitudes may limit its usefulness at the global scale (Duguay et al., 2015). However, the integration of optical imagery (e.g., MODIS, Landsat, Sentinel-2) and radar altimetry data provides long-term continuity in the production of consistent and calibrated records, and we encourage to re-explore the lakes in our study using Landsat and/or Sentinel images with 20-30m spatial resolution."

**2. How many lakes in the Tibetan Plateau are included? The existed studies have reported that about 60 lakes with altimetry data and the corresponding estimates of lake volume variations.**
We created elevation, area, and storage variation records for 30 lakes in the Tibetan Plateau (cfr. list below). As explained in the previous comment, the spatial resolution of the satellite imagery limited the number of lakes for which a reliable WSE-WSA relationship could be established to estimate storage variation. We now highlight this in page 9, line 5-7 as reported below.
"The majority of the water bodies (223, 64.26% of the total) are located in Asia (110, of which 30 in the Tibetan Plateau) and North America (113), with Australasia represented by just eight targets."

**3. The Equation (1) is correct? It is WSEt+1 - WSEt?**
We thank the reviewer for spotting the typo. We edited Equation (1) in the manuscript as:
$\Delta V = (WSA_{t+1} + WSA_t)(WSE_{t+1} - WSE_t)/2$

**4. The linear regression between elevation (WSE) and surface area (WSA) was used. How about polynomial correlation? The authors test them?**
We agree with the reviewer's comment that the bathymetry of a lake should not follow a linear behavior, and acknowledge that the 0.5 multiplier used in Equation (1) usually underestimates the actual volume change by not taking into account factors such

non-linear bathymetries and the shape of the shoreline. However, volume changes are dominated by WSE rather than WSA changes, suggesting that bathymetry errors are less important than WSE errors. Such approach works reasonably well at most lakes/reservoirs (cfr. Gao et al., 2012), ultimately proving more portable to lakes/reservoirs at the global scale. We now account for this in page 15, line 25-35 as reported below.

"The quality of both elevation and surface area contribute to the accuracy of their relationship, but volume changes are mostly dominated by elevation changes. High correlations between elevation and area generally indicate reliable $\Delta$V estimation. However, if either variable is systematically biased, the error associated with the relationship is carried to the estimated $\Delta$V. For example, low correlation may arise when the target shows nearly constant WSA (vertical walls, in which case a variation in elevation reflects in a negligible change in WSA) or nearly constant elevation (i.e., shallow lakes, in which case a variation in surface area reflects in a negligible change in elevation). In these cases we proceeded in the modelling of $\Delta$V with the parameterization of the invariant variable with its mean value. All the factors listed above introduce some degree of error in the WSE-WSA relationship; however, in most cases a linear approximation does not appear to be a major contributor (cfr. Gao et al., 2012)."

**5. More validations including lakes in different types and continents can be provided?**

We thank the reviewer for the recommendation and we agree that validation using further lakes would be beneficial to the global nature of our study. However, we state how "[the] records presented in this paper represent the most complete satellite-derived global surface water storage time series to date, spanning from 1992 (TOPEX-Poseidon launch) to present, with the potential to be extended up to the launch of the SWOT mission planned for 2021" (page 16, line 31-34). In addition, we acknowledge that "[the] data set presented is dynamic and will continue to be extended both in terms of the number of water bodies (with ultimate potential total around 400), and historical time period" (page 16, line 34-35), but this is beyond of the scope of the manuscript.

**6. How water storage change in 1992-2000 without MODIS water mapping was estimated?**

As explained in section 2 Data and methods (page 7, line 27-28), we used linear regression to approximate the relationship between WSE and MODIS-derived WSA when concurrent measurements are available (2000-2016), and then applied this relationship to estimate WSA from WSE for periods when WSA is unavailable (1992-1999).

**Specific comments:**

**1. "and to characterize how these conditions change through time over long periods (Lettenmaier et al., 2015; Crétaux et al., 2016)" A suggested study here for monitoring lake area, level and volume changes since 1970s: http://dx.doi.org/10.1002/2017GL073773**

We thank the reviewer for the recommendation and added the reference to the text as suggested. Zhang G., Yao T., Shum C. K., Yi S., Yang K., Xie H., Feng W., Bolch T., Wang L., Behrangi A., Zhang H., Wang W., Xiang Y., and Yu J.: Lake volume and groundwater storage variations in Tibetan Plateau's endorheic basin. Geophys. Res. Lett., 44(11), 5550-5560, https://doi.org/10.1002/2017GL073773, 2017.

**2. "and the Tibetan Plateau (Lee et al., 2011; Kleinherenbrink et al., 2015; Cai et al., 2016)." It looks some key studies of lake changes in the Tibetan Plateau from altimetry data are missed here. https://doi.org/10.1029/2019GL085032, https://doi.org/10.1016/j.rse.2011.03.005, https://doi.org/10.1007/s10712-016-9362-6.**

We thank the reviewer for the recommendation and added the references to the text as suggested.

Zhang G., Xie H., Kang S., Yi D., and Ackley S. F.: Monitoring lake level changes on the Tibetan Plateau using ICESat altimetry data (2003-2009). Remote Sens. Environ.,

115(7), 1733-1742, https://doi.org/10.1016/j.rse.2011.03.005, 2011.

Crétaux J. F., Abarca-del-Río R., Bergé-Nguyen M., Arsen A., Drolon V., Clos G., and Maisongrande P.: Lake Volume Monitoring from Space. Surv. Geophys., 37(2), 269-305, https://doi.org/10.1007/s10712-016-9362-6, 2016.

Zhang G., Chen W., and Xie H.: Tibetan Plateau's Lake Level and Volume Changes From NASA's ICESat/ICESat‐2 and Landsat Missions. Geophys. Res. Lett., 46(22), 13107-13118, https://doi.org/10.1029/2019GL085032, 2019.

**3. "a polygon was drawn by hand using high resolution imagery from various sources (e.g., Global Surface Water Explorer, Google Earth, ESRI World Map)" How to make sure the dates between them are matched.**
The reviewer brings up a valid point here. However, as explained in section "2.2 Surface water area", these polygons are exclusively used as initial reference for the classification and water surface area extraction. Given the nature of the classification algorithm described in Khandelwal et al (2017), mismatches between actual water surface area extent and reference polygons are taken into account by introducing the concept of "dynamic region width" (page 15, line 17-26). Ultimately, the water surface area records in the GOLA data set are exclusively a function of the MODIS imagery utilized, but volume changes are mostly dominated by elevation changes. We clarify this aspect in the manuscript as follows (page 15, line 21-26).
"Due to the moderate spatial resolution of the GOLA records, the effect of mixed pixels is even more prominent in water bodies with low dynamic region width, which can lead to low correlation values between elevation and surface area. Conversely, the classification of targets with high dynamic region width consistently performs better in the GOLA records. The quality of both elevation and surface area contribute to the accuracy of their relationship, but volume changes are mostly dominated by elevation changes."

**4. The links below are not accessed. https://doi.org/10.5067/UCLRS-GREV2,**

**https://doi.org/10.5067/UCLRS-AREV2, https://doi.org/10.5067/UCLRS-STOV2, https://podaac.jpl.nasa.gov/**
The links listed provide the location of the data repositories, and they are all active and publicly accessible. We now state this explicitly in page 16, line 17-18.
"The links listed provide the location of the data repositories, and they are all active and publicly accessible."

**5. The correlation (r2) could be presented in Figures 6, 7, 8.**
We agree with the reviewer that adding the R2 would enhance the figures' readability. We have done so where applicable (Figure 6 and 8) as suggested. Figure 7 is instead limited to the 2000-2016 period where both WSE and WSA were used to calculate the hypsometric function used to extrapolate records pre-2000 and post-2016.

**6. How about the mismatching in Figure 8a-b?**
As explained in the text (page 12, line 1-7), we evaluated the statistical accuracy of WSE and storage estimates at Lake Sakakawea based on monthly in situ water measurements at Garrison Dam (black). These measurements were plotted against (a) monthly average WSE records and (b) storage change estimates (red), resulting in 233 and 270 coincident observations, respectively. Panel (c) and (d) directly compare the correspondent measurements, and as described in the text (page 12, line 8-11) the linear fits resulted in R2 0.95 and 0.94, respectively, indicating very good consistency with the in situ measurements.

---

## Author Comment (AC2) · 27 Mar 2020

We are thankful to the anonymous reviewer for the thoughtful comments, which led to significant improvements to our manuscript. All comments were addressed as described below.

Best regards,
Riccardo Tortini

The manuscript entitled "Satellite-based remote sensing data set of global surface water storage change from 1992 to 2018" by Tortini et al. presents estimated global surface water storage changes ( $\triangle V$ ) in large lakes and reservoirs using a combination of paired water surface elevation (WSE) and water surface area (WSA) extent products. In their approach, they used data produced by multiple satellite altimetry missions (TOPEX-Poseidon, Jason-1, Jason-2, Jason-3, and ENVISAT) from 1992 on, with surface extent estimated from Terra/Aqua Moderate Resolution Imaging Spectroradiometer (MODIS) from 2000 on. They used the relationships between elevation and surface area to produce estimates of $\triangle V$ even during periods when either of the variables was not available. They produce time series of $\triangle V$ as well as WSE and WSA for a set of 347 lakes and reservoirs globally for the 1992-2018 period. In general I find the idea of manuscript very interesting and I also see the need for having such data base. Indeed, the production of long-term, consistent, and calibrated records of surface water cycle variables such as the data set presented here is of fundamental importance to baseline future SWOT products.
We thank the anonymous reviewer for the kind words.

**Major comments:**
I believe the paper suffers from missing an important point. The authors calculate first the correlation coefficient between the two variables and use it as one of the decision parameter for taking the mean value instead of data itself. The correlation coefficient between two data set represents the linear dependency between two data sets, while the relationship between water level and surface area represent the bathymetry and the bathymetry of a lake should not follow a linear behaviour. I would strongly suggest to change this in the paper and in case the authors would like to assess the monotonic behaviour between water level and area, then they should use the Spearman rank correlation and not simply the Pearson correlation.

We agree with the reviewer's comment that the bathymetry of a lake should not follow a linear behavior, and acknowledge that the 0.5 multiplier used in Equation (1) usually underestimates the actual volume change by not taking into account factors such non-linear bathymetries and the shape of the shoreline. However, such approach works reasonably well at most lakes/reservoirs (cfr. Gao et al., 2012), ultimately proving more portable to lakes/reservoirs at the global scale. We now account for this as explained in page 15, line 25-35.

"The quality of both elevation and surface area contribute to the accuracy of their relationship, but volume changes are mostly dominated by elevation changes. High correlations between elevation and area generally indicate reliable $\Delta V$ estimation. However, if either variable is systematically biased, the error associated with the relationship is carried to the estimated $\Delta V$. For example, low correlation may arise when the target shows nearly constant WSA (vertical walls, in which case a variation in elevation reflects in a negligible change in WSA) or nearly constant elevation (i.e., shallow lakes, in which case a variation in surface area reflects in a negligible change in elevation). In these cases we proceeded in the modelling of $\Delta V$ with the parameterization of the invariant variable with its mean value. All the factors listed above introduce some degree of error in the WSE-WSA relationship; however, in most cases a linear approximation does not appear to be a major contributor (cfr. Gao et al., 2012)."

**My second major comment goes to the methodology for the area extraction. Figure 6 shows some vertical lines of points, which represent same area for different water levels. This is highly suspicious.**
We acknowledge that the area classification algorithm may suffer from uncertainty due to the spatial resolution of the imagery used (i.e. 500 m). However, using MODIS imagery over other finer resolution satellite images (e.g. Landsat) ensured to obtain a denser observation time series (virtually 32 times higher) due to the satellite revisit times (i.e. two daily MODIS observations vs Landsat's 16-day revisit time. This ultimately led to establishing a more robust relationship between WSE and WSA in
order to model ∆V. We now emphasize this point in page 16, line 1-10 as reported below.

"Despite GOLA's moderate spatial resolution it can potentially affect the accuracy of ∆V estimates, higher resolution satellite missions have longer satellite revisit time (e.g., 16 days for Landsat, 10 days for Sentinel-2A starting in 2015 and 5 days for Sentinel-2A and -2B in tandem formation starting in 2017). Because we leveraged the relationship between WSE and WSA to estimate ∆V, such satellite revisit times would produce sparser records, especially for water bodies located at high latitudes and/or altitudes as they are more affected by cloud cover. In fact, despite being highly desirable for monitoring of surface water dynamics, imagery from optical sensors is strongly affected by the presence of cloud cover, which can be extensive in late fall and winter, and in combination with low sun angle experienced at high latitudes may limit its usefulness at the global scale (Duguay et al., 2015). However, the integration of optical imagery (e.g., MODIS, Landsat, Sentinel-2) and radar altimetry data provides long-term continuity in the production of consistent and calibrated records, and we encourage to re-explore the lakes in our study using Landsat and/or Sentinel images with 20-30m spatial resolution."

**Specific comments:**
**page 2, line 30, please mention River and Lake https://earth.esa.int/web/guest/-/river-and-lake-products-from-altimetry-4617 and HydroSat http://hydrosat.gis.unistuttgart.de**
We thank the reviewer for the suggestion. We now mention both in page 2, line 34-37.
"Further examples of global data sets are the University of Stuttgart's HydroSat (http://hydrosat.gis.uni-stuttgart.de/; accessed February 27th, 2020), and, despite being no longer actively maintained, the European Space Agency's River Lake Altimetry products (http://altimetry.esa.int/riverlake; accessed February 27th, 2020)."

**Page 4, Section 2.1, Did you make sure that all data from different data**

**centers have the same background models for atmospheric refraction? How did you deal with the intersatellite bias?**

The reviewer brings up an important point here. The G-REALM10 products are constructed from the merger of (up to) four mission datasets. Elevation reconstruction within each mission is based not only on the version (standard) of the data set but also what atmospheric and tidal corrections are currently available for that mission. The 10-day products (G-REALM10) are thus a blend of Geophysical and Interim Geophysical data records, and a mix of data version's B through D. The G-REALM35 products (based on the ENVISAT mission) are dataset version 2.0. The atmospheric range corrections also vary between the datasets, for example, while the radiometer based wet tropospheric range correction is the primary selection across all, the secondary model-based choice utilizes the ECMWF estimates for Jason-2, Jason-3, and ENVISAT but is currently limited to employing the RADS/ERA model correction (TOPEX/Poseidon, Jason-1). The ionospheric range correction can also differ between missions, e.g., selecting the GIM model (Jason-3, ENVISAT) but otherwise utilizing various RADS options (TOPEX/Poseidon, Jason-1, and Jason-2). Full processing details can be found in the project ATBD document for the lake level products (Birkett et al., 2019). The altimetric community continues to upgrade mission datasets and is striving for a more common dataset version/standard across all missions.

Merging (up to 4) 10-day resolution time series to create one uniform product spanning multiple decades relies on the availability of data within the 6-month overlap periods i.e., when the historical and new mission are in a tandem phase, overpassing the lake on the same ground track but spaced 1 minute apart. Any inter-mission range bias can be corrected for by noting the elevation shift required to align the results from the two time-series. Absence of data in this overlap period results in the application of a global mean inter-mission range bias estimated from global observation of ocean surface heights (Birkett et al., 2019, ATBD document). Merging a combination of GREALM-10, GREALM-35, DAHITI or LEGOS products to obtain the longest time record also cannot ensure uniformity of atmospheric corrections across all the different product

sources. In these merger cases elevation bias were estimated from the difference of the means of a subset (with good periodicity and few outliers) of each series.

We now emphasize this point in page 4, line 21-22 and page 5, line 1-2.

"Full details of the processing to create the G-REALM10 and G-REALM35 products can be found in the Algorithm Theoretical Basis Document (ATBD; Birkett et al., 2019). This includes the descriptions of the atmospheric corrections applied in the height reconstructions, the inter-mission height bias application, and the inherent differences between mission data set versions."

In addition, we now reference the data sets' respective ATBD in "Abstract" (page 1, line 27) and "6. Data availability" (page 16, line 15).

"The data sets presented and their respective Algorithm Theoretical Basis Documents are publicly available and distributed via NASA's Jet Propulsion Laboratory's Physical Oceanography Distributed Active Archive Center (PO DAAC; https://podaac.jpl.nasa.gov/)."

**page 5 line 3, what is an acceptable accuracy? please quantify!**

Our approach utilizes the classification algorithm described in Khandelwal et al. (2017) to estimate water surface area from MODIS imagery. In their paper, the authors validate the MODIS-based classification maps (500 m resolution) using higher spatial resolution Landsat-based reference maps (30 m resolution) at three target reservoirs (Mead, Kremenshugskoye, and Nova Ponte) under a dry and wet scenario (cfr. Table 2 in Khandelwal et al., 2017), discussing potential and limitations of such approach. Given the global nature of our study, it is virtually impossible to single-handedly establish "an acceptable accuracy" for 347 lakes/reservoirs.

**page 7, equation 1, I did not grasp the equation. shouldn't be WSA t+1 - WSA t?**

We thank the reviewer for spotting the typo. We edited Equation (1) in the manuscript

as:
$\Delta V = (WSA_{t+1} + WSA_t)(WSE_{t+1} - WSE_t)/2$

**page 7, line 26, considering linear regression is wrong. See my major comment.**
We are thankful to the reviewer for further reinforcing this point, addressed in the reply to the major comment above.

**Figure 6, the extracted area is so noisy that similar area are obtained for different height. And in fact, no obvious linear relationship can be recognized.**
We thank the reviewer for further highlighting this point, due to the resolution of the MODIS imagery utilized as discussed in previous comments.

———————————————————

---

## Author Response (AR2)

Dear Editor,

We are thankful for further underlying the importance of the point highlighted by the anonymous reviewer on the statistical method applied, which has to do with appropriateness of Pearson v. nonparametric (e.g., Spearman) correlation estimates.

Many studies have been reported in the applied statistics literature on the appropriateness of least squares vs. nonparametric estimators (e.g., of correlation; Pearson vs. Spearman, Kendall's Tau, or others). The general finding is that parametric estimators perform slightly better than nonparametric ones so long as the underlying distribution(s) are "well behaved" (i.e., don't have heavy tails, for instance normal) but perform much worse in the presence of heavy tails. So the question of appropriateness of Pearson's correlation in our case really comes down to whether the data we analyzed have heavy tails. As described in the ATBDs accompanying each data set (which we cite), we ensured the absence of outliers in the WSE and WSA products. Specifically, each product was filtered using a 30-day median window to reject potential outliers. At the selected example site (Lake Sakakawea), WSE and WSA show a strong linear correlation with $R^2 = 0.908$ (Figure 6), which confirms the appropriateness of using a parametric (Pearson's) correlation coefficient. Furthermore, by providing both the WSE and WSA products, it is possible for users to apply any statistical method they desire based on their needs in order to estimate storage change.

For these reasons we argue that the statistical methods we have used are appropriate to the data products described in the paper.

Should you have any further concern, please do not hesitate to contact me.

Sincerely,
Riccardo Tortini